# Psychological First Aid Training: A Scoping Review of Its Application, Outcomes and Implementation

**DOI:** 10.3390/ijerph18094594

**Published:** 2021-04-26

**Authors:** Ling Wang, Ian Norman, Tao Xiao, Yamin Li, Mary Leamy

**Affiliations:** 1Clinical Nursing Teaching and Research Section, The Second Xiangya Hospital, Central South University, Changsha 410011, China; aminny@csu.edu.cn; 2Florence Nightingale Faculty of Nursing, Midwifery & Palliative Care, King’s College London, London SE1 8SW, UK; ian.j.norman@kcl.ac.uk (I.N.); mary.leamy@kcl.ac.uk (M.L.); 3Department of Orthopedics, The Second Xiangya Hospital, Central South University, Changsha 410011, China; xiaotaoxyl@csu.edu.cn

**Keywords:** Psychological First Aid, PFA training, psychosocial response, early psychosocial support, mental health and psychosocial support, non-specialist, healthcare workers, mental health preparedness, mental and behavior health management, capacity building

## Abstract

Worldwide commitment to disseminate Psychological First Aid (PFA) training to enable frontline workers to support distressed individuals and/or manage their own self-care is increasing, but the evidence base of PFA training is uncertain. Method: a scoping review was undertaken by searching seven databases and hand-searching grey literature to maximise coverage of potential studies. Results: Twenty-three studies met the inclusion criteria. Three PFA training models were commonly used in research studies. A broad selection of PFA training outcomes were observed including learning, behavior, satisfaction and practice in crisis mental and behavior health preparedness. Conclusions: Research evidence of reasonable quality demonstrates that PFA training significantly improves knowledge of appropriate psychosocial response and PFA skills in supporting people in acute distress, thereby enhancing self-efficacy and promoting resilience. However, this review highlights inadequate guidance on how PFA training should be applied and adapted, significant shortcomings of reporting PFA training delivery, limited training evaluation and unclear training outcomes. Whilst behavioral, knowledge and system impact of the PFA training are promising, methodologically stronger evaluations which include systematic training adaptation and selection of sensitive outcome measures is needed to strengthen future implementation of PFA training and thereby enhance population preparedness for future emergencies.

## 1. Introduction

At time of writing, the mental health crisis posed by the COVID-19 pandemic has reached an epidemic level, threatening the health and economic prosperity of all nations [1]. Early psychosocial response to those affected by such traumatic events is necessary to prevent the onset of mental illness [2]. Frontline healthcare workers are at the forefront of supporting acutely distressed individuals through providing early psychosocial care. Apart from their pivotal role and responsibility for patient care, frontline health workers are also at high-risk themselves of moral injury and mental health problems arising from working in overwhelming stressful situations. This can, in turn, have negative consequences for the quality of patient care and increased attrition rates in the health professions [3,4]. This situation is considered by some as a consequence of an historical underinvestment in mental health prevention, especially in preparing healthcare workers to manage their own self-care [5].

The COVID-19 pandemic is the latest crisis to highlight the importance of Psychological First Aid (PFA) training as being critical to empower frontline workers to mitigate the adverse impact of crises through primarily supporting people in need and promoting self-care [6,7]. However, PFA as a frontline approach to providing immediate psychosocial care is not new. First coined at the end of World War II PFA became increasingly popular in the 1980s and 1990s amongst disaster mental health experts since it offered an integrated continuum of critical incident stress management techniques [8,9]. Despite its long history, PFA has emerged as a mainstay for early psychological intervention only since the post-9/11 era [10,11]. Increasing concern about the unintended consequences of psychological debriefing (e.g., increased risk of PTSD and depression) has accelerated the use of PFA as a “do no harm” alternative and gained a broad professional consensus [11,12]. Since then, numerous PFA frameworks and models have been developed emphasising different components and areas of focus for specific populations and organisations [11]. Whilst here is no universally accepted definition of PFA across different models, the five Hobfoll’s intervention principles established in 2007 has exerted a profound influence on the PFA as an emerging field [13]. For instance, World Health Organization which published its PFA guide in 2011 [14] and the Johns Hopkins PFA Guide published in 2017 [15] both incorporate essential elements of other previous PFA models to assist recovery and both are underpinned by five intervention principles which aim to promote: a sense of safety, calming, a sense of self and community efficacy, connectedness and hope. 

While PFA as a therapeutic intervention is still lacking sufficient evidence to support a clinical treatment standard or guideline [16,17,18], this does not, however, detract from the potential utility of PFA training for frontline health professionals. PFA training is an educational intervention that aims to equip workers with the knowledge and skills to deliver PFA as the therapeutic intervention to individuals who require immediate psychological support following a crisis. PFA training as an educational intervention to ensure universal crisis mental health preparedness has several potential advantages. Firstly, PFA as a therapeutic intervention does not involve diagnosis and treatment but provides health workers with a stepwise approach to delivering mental health support which includes approaching the person non-judgmentally, assessing their needs, supporting them with information gathering and linking them with further resources [14,19,20]. Second, as an entry point to more intensive mental health skills, PFA training offers health care professionals with expertise in delivering a low-intensity, step-by-step, therapeutic approach which can be applied in a wide range of humanitarian and disaster situations [10]. Given this, PFA training seems well-suited to delivery within the curriculum for non-specialist frontline workers to prepare them to provide psychosocial care for distressed individuals whilst also ensuring a resilient workforce able to respond effectively to disasters and emergencies [21,22].

The last decade has witnessed a boom in various PFA capacity building and training initiatives for disaster mental health preparedness. Recognizing the significance in mounting psychosocial support following critical events, the dissemination of PFA training has evolved into a frontline psychosocial response skillset that now govern and define how we conceptualize mental health crisis response [23,24]. It is widely accepted that PFA training can be adapted to meet the priorities of the trainees and/or their practice context and delivered on different platforms (live, online and mobile), so creating numerous options for the training programmes [25,26,27]. Furthermore, the increasing availability and promotion of PFA training reflects increasing recognition of the need for mental health crisis mitigation at a population level. For example, PFA was identified as the theme of 2016 World Mental Health Day which aimed to improve mental awareness and called for broad PFA training reach [27]. Advocates of PFA training propose that it should be accessible to all healthcare workers to increase their coping ability, reduce their risk of distress and provide them with skills to better deliver person-centred care [28,29,30]. However, in spite of increasing opportunities for PFA training the current state of PFA training and the outcomes of training for trainees and ultimately for disaster victims are uncertain. 

Since the COVID-19 pandemic is likely stimulate increased interest in PFA training to equip healthcare workers to manage the mental health impact of any future pandemics, it is more important than ever to map the existing literature on PFA training The purpose of this review is to examine PFA training evidence to establish the breadth of current training applications, its implementation and outcomes, as well as the barriers and enablers to training in order to guide future PFA training use and support the potential scaling up frontline capacity-building approach.

## 2. Materials and Methods

Given the broad purpose, a scoping review was considered most appropriate since it provides a systematic methodology for determining the state of the evidence on a topic. Therefore, this review followed the guidelines proposed by Arksey & O’Malley [31], which involves five key steps: developing the research question; identifying relevant studies; selecting studies according to inclusion and exclusion criteria; charting the data; collating, summarising and reporting the results and considering the implications of the study findings for policy, practice and research [32]. 

### 2.1. Identifying the Research Question

The research question was identified as: ‘What is the current application of PFA training as an evidence based educational intervention? To address the research question, the following two objectives were developed: (a) To identify and describe the key components of PFA training and its implementation; and (b) To summarise the evidence on the implementation and outcomes of PFA training.

### 2.2. Identifying Relevant Studies

Considering the broad uncertainties on its use and various terms used to describe PFA, a pilot search was conducted to establish a comprehensive background understanding of PFA. Informed by a previous review of PFA [11], all related search terms including “psychological first aid”, “PFA”, “community-based psychosocial support”, “disaster behavior first aid” and “stress first aid” were included in the pilot search. In this pilot search, no methodological, language or date restrictions were employed. The search results were screened by the first author and discussed with the co-authors. 

The pilot search revealed that studies with PFA training as an educational intervention did not always include PFA in the title or abstract, did not always include or included only in part the five key elements—and so a search strategy using the words “Psychological First Aid” and “PFA” would be sensitive and would identify a very high proportion of relevant literature. Given this, the review adopted a two-step approach. The approach taken was to: (1) use a variety of terms searching PFA, and then; (2) hand search the papers identified in step one to identify training intervention studies.

The comprehensive search was conducted within seven electronic databases (MEDLINE, PsychINFO, CINAL, Cochrane, Web of Science, PILOTS and CNKI (Chinese) since 1945, which was the year that the term “psychological first aid” was coined in American medical literature by Blain, Hoch and Ryan [8]. The refined search strategy for the literature involved: (i) a traditional search: as an example, the MEDLINE database search for “Psychological First Aid” was (Psychological First Aid *).mp. (mp = title, abstract, original title, name of substance word, subject heading word, keyword heading word, protocol supplementary concept word, rare disease supplementary concept word, unique identifier); (ii) use of internet search engines; and (iii) grey literature by hand-searching reference lists (for example in relevant reviews or journal papers), by using a snowballing process by searching the internet (through Google and Google Scholar search engines, relevant organisational websites and key journals) [33]. 

### 2.3. Study Selection 

The inclusion and exclusion criteria (see Table 1) were developed based on the PICOS (population, intervention, comparisons, outcomes, study) framework [34,35].

### 2.4. Charting of Information and Data within the Included Studies

After removal of duplicates, titles/abstracts were screened in duplicate using Covidence software [36]. Disagreements and uncertainties were resolved by discussion with reference to the title and abstract. Full-text articles were retrieved if inclusion criteria were met or if the abstract did not contain sufficient information; then screened in duplicate with disagreements resolved by discussion. 

Data exaction tables were created approved by all authors that included: details of different PFA models comparison (Table 2); general study characteristics -author, year, country of study, aim, design, outcome and key findings (Table 3); summary of current training and outcomes using the Template for Intervention Description and Replication (TIDieR) checklist (Table 4); outcome measures of the training (Table 5). 

A formal quality assessment of the articles was not performed as this was not required in line with the scoping review methodology and the paucity of PFA training effectiveness evidence.

### 2.5. Collating, Summarising and Reporting Results

An analytical descriptive method was used to chart the data on the PFA training intervention and to extract contextual or process-oriented information from included studies [37]. Studies were charted according to the TIDieR checklist items, e.g., underlying theory (why), materials and procedures used (what), the intervention provider (who), the mechanism of delivery (how), the location (where), intervention schedule and intensity (when and how much), whether adaptations were used (tailoring) and the intervention adherence (how well) [38]. A narrative synthesis approach was used to provide a thorough account of the data and to draw comparisons across different PFA training intervention.

## 3. Result

### 3.1. Study Selection

The initial search returned 2592 records (Figure 1). Screening of titles and abstracts excluded 1362 articles, leaving 236 as potentially eligible. Screening of full-text items excluded an additional 213 articles, for the following reasons: lack of model description (*n* = 19), no outcome reported (*n* = 77), ineligible publication type (*n* = 88), not English/Chinese language (*n* = 16) and not PFA training (*n* = 14). Of the remaining 23 eligible articles 20 used a quantitative research design and three used a qualitative research design.

### 3.2. PFA Models 

The most commonly used PFA models in training evaluations studies were the Psychological First Aid Guide for Field Workers [14], the Psychological First Aid: Field Operations Guide [20], the Johns Hopkins Guide to Psychological First Aid [15]. A comparison table of these PFA models and the Immediate Cognitive-Functional Psychological First Aid (ICF-PFA) model (which is used less frequently), is provided below (see Table 2). This table compares models using Hobfoll’s five principle goals of PFA: (i) how to approach and listen, (ii) assess the needs, (iii) provide support with information, (iv) referral and (v) self-care. The available resources and settings in which evaluations have been undertaken are also compared.

#### 3.2.1. Psychological First Aid: Guide for Field Workers

This model was jointly developed by the World Health Organisation, World Vision International and War Trauma Foundation based on the contributions of 60 international peer reviewers and has been endorsed by 24 humanitarian organisations. The guide’s authors deliberately used simple language to make it easily adaptable for diverse socio-cultural settings within in low-and middle-income countries (LMICs). As PFA was part of a broader response to crises which was recommended by the Inter-Agency Standing Committee guidelines on Mental Health and Psychosocial Support (MHPSS), it fueled the dissemination of the WHO PFA guide in humanitarian aid settings. The PFA guide for field workers was subsequently developed and accompanied by a facilitator’s manual and tailored for specific fields, for example, Ebola Virus Disease, and has been translated into more than 20 languages. In order to drive the humanitarian aid capacity-building initiatives mostly in LMICs, WHO, in collaboration with international organisations and local institutions offered various resources and networks to facilitate the spread of PFA training courses worldwide [39,40,41,42].

#### 3.2.2. Psychological First Aid: Field Operations Guide

This model was created collaboratively between the National Child Traumatic Stress Network (NCTSN) and the National Center for Posttraumatic Stress Disorder in America. The National Center for PTSD, the International Federation of the Red Cross (IFRC) and the American Red Cross (ARC), had all advocated that PFA training should be a mandatory, priority skills-based training for first responders, in order to equip them with the skills and knowledge to address the mental health consequences of responding to disasters and public health emergencies. Since the second and most current edition of the guide, published in 2006, several translations and system-specific adaptations of the PFA guide along with the training approaches had been developed and are available as open access resources [20]. The distribution of operation manuals for related groups (such as homeless, nursing home staff, medical reserve corps volunteers) had explicitly contributed to wide-spread application of PFA as a disaster intervention tool [43,44,45,46,47]. 

#### 3.2.3. The Johns Hopkins Guide to Psychological First Aid

This model was developed by George Everly and Jeffery Lating. It was based upon their accumulated field experience, whereas the guide was produced in collaboration with the Johns Hopkins University’s, United States Centers for Disease Control and Prevention, who provided organisational support. As one of the most disaster-prone countries aiming to enhance surge capacity, a large initiative representing 14 preparedness and emergency response learning centres were established to equip the public health workforce with PFA competencies which further shaped the present model through expert consensus. In contrast to the first two models, the Johns Hopkins Guide, (also known and referred to as the RAPID PFA model), purports to be a theory-driven, evidence-informed and empirically validated model [48,49]. A therapeutic intervention, based on the Johns Hopkins Guide to PFA has been evaluated in a randomised controlled trial [50,51].

#### 3.2.4. Mixed Models

A blend of the elements of the three commonly used models described above was also identified. These modified PFA training programmes were tailored for delivery to college students and school counsellors [52,53,54]. Amongst these mixed PFA models was the SIX Cs model (also known as immediate cognitive-functional PFA ICF-PFA), which contained six basic elements: cognitive communication, challenge and control, commitment and continuity [55]. 

### 3.3. Characteristics of Included Studies

PFA training evaluations (Table 3) have been undertaken in 10 countries: Haiti, Israel, Turkey, Korea, Japan, China, Sierra Leone, Lebanon, Australia and the USA. Of these, 15 studies were conducted in upper-and-middle income countries (*n* = 15, 68%), most notably in the USA (*n* = 11, 52%). The majority (*n* = 19, 83%) of the PFA training courses were applied by humanitarian aid and disaster response sectors and mostly delivered to non-specialists first responders such as Non-Government Organisation (NGO) staff, health workers, lay community citizens, faith leaders and police. The sample sizes ranged from 76 to 1500 (Mean = 82). Most PFA trainees were female (i.e., 65% to 99%). Of six studies reporting the length of trainees work experience, half of the school counsellors and medical reserve corps had less than 5 years in employment [46,53], however, nursing home staff and police trainees had more than 14 years’ work experience [45,56]. More than 50% of trainees were educated to degree level in a related discipline (e.g., psychology, social work, nursing, education) [44,46,57], and among primary healthcare workers responding to Ebola outbreak less than 10% were educated to diploma level [40]. More than 50% trainees (community leaders and nursing students) had no prior disaster or emergency experience [52,56]. 

Most studies used a pre-and-post intervention research design (*n* = 14, see Table 3), whereas four studies only collected a post-intervention data. Even though four studies conducted a randomised controlled trial (RCT) [42,47,52,54], their randomisation methods were not described in detail. In the seven studies that conducted follow-up evaluations, these ranged from two weeks to six months follow up periods [42,44,47,52,55,56,57]. Six studies reported using a mixed-method evaluation design and three studies only collected qualitative data. Few studies collected views of third parties within their evaluations (e.g., course facilitator’s field notes, subjected trainees to observation by ward mangers or collected data from service users) [56,57,58]. The selection and type of outcomes measures varied enormously across studies (Table 5).

### 3.4. The Characteristics of PFA Training Delivery 

Table 4 presents results using the TIDieR [38] reporting checklist to describe the content and delivery of PFA training interventions, such as “why”, “who”, “how”, “when” and “how much”, “tailoring” and “how well” for the 20 quantitative studies. The table does not include the three qualitative studies as they failed to report any details of the PFA training. 

#### 3.4.1. Theory (Why?)

All studies reported either a rationale, a goal, or an underpinning theory of the PFA training intervention with regard to training design and/or delivery. Most studies provided a rationale statement for the training, whereas two studies also incorporated theoretical concepts. Everly et al. justified the theoretical foundations for the Johns Hopkins PFA model, indicating the need to address determinants of environmental stressors via cognitive and subsequent affective change to establish the eight learning objectives in the training intervention [48]. Farchi stated four theoretical concepts of the ICF-PFA model: hardiness; sense of coherence; self-efficacy; and neuropsychology of the stress response [55].

#### 3.4.2. Material and Procedures (What?)

Most studies reported their training intervention procedures or blended materials (*n* = 19, see Table 4). A few studies provided a full description of the training content covering theoretical knowledge (disaster mental health, resilience, common stress reactions) and subsequently PFA knowledge (basic principles, core actions of PFA, self-care). An educational package as a disaster toolkit was occasionally assembled for the trainees which incorporated PowerPoint slides and educational material about disaster mental health (e.g., literature references, videos, posters and brochures) [58,62]. Given the subsequent adaptations of the training in each study, no studies provided details of where their new materials could be accessed (e.g., online appendix, URL). 

#### 3.4.3. Training Provider (Who?)

PFA training courses were provided by mental health professionals, namely three psychiatrists [48,49,62], four psychologists [47,53,54,63] and five mental health researchers or practitioners [41,42,43,52,61]. Non-mental health professionals also provided the PFA training such as trained facilitators [44,58] and trained health workers [45,56].

#### 3.4.4. Mechanism of Delivery and Location (How and Where?)

Current training evaluation studies had been done all using face-to-face sessions delivered in the workplace. These included interactive didactic presentations (*n* = 17) and role-play or simulation practice (*n* = 9) [42,44,46,47,49,52,53,57,63]. A few studies reported providing a train-the-trainer workshop (*n* = 4) [42,45,46,61]. The PFA training was administered for groups but only three studies stated the group size: 15–20 Non-Government Organisation (NGO) workers [41], 9–10 students in Kilic study [52] and three public health workers in role-playing reflective listening [49]. Of note, only Sijbrandij et al. reported providing supervision after the training course [42].

#### 3.4.5. Training Schedule and Intensity (When and How Much?)

The length of PFA training courses varied considerably, ranging from a minimum of 90 min to a maximum of 6 days. Most PFA training was given in regular sequential sessions (e.g., spread between 1–2.5 days), with the majority in a one-day course (7–8 h) (*n* = 11, 55%). For example, the RAPID-PFA training programme was delivered in a 6-h/session in a one-day format. Some training interventions were provided in a very short format due to time constraints in real settings [42,45,61]. Most studies did not report full details of training intensity. Of those that did eight training courses were of a short-to-medium duration ranging from 4 session to 20 sessions and five studies reported that the training was delivered over three or four sessions [56,58,61,62,64].

#### 3.4.6. Adaptations (Tailoring)

Most of the training courses incorporated some adaptation, although few elaborated upon the procedure for these adaptations. None of the studies incorporated systematic approaches to adaptation for example, stakeholder involvement to programme design, or testing and refinement of the training intervention. The PFA training content was tailored to the specific population or contexts, for instance, a health assistant-oriented version included problem solving, relaxation, breathing and mindfulness techniques, sleep hygiene education, stress management, social mobilisations, behavioral activation and self-care strategies [57]. 

#### 3.4.7. Adherence and Fidelity (How Well)

The fidelity check of the training was only supported by three studies [42,53,58]. Only Sijbrandij et al. reported 72% to 95% adherence to key features of PFA training [42].

### 3.5. Outcomes of PFA Training

We have used Kirkpatrick’s levels of training evaluation to classify the outcomes into four subthemes: change in practice (evaluation of behaviour), change in knowledge, skills, attitude (evaluation of learning), satisfaction with training (programme evaluation) and change in participant outcomes (evaluation of trainee practice from recipients or observers) [65] (see Table 5).

#### 3.5.1. Effect on Learning

Of the 14 studies reporting successful learning outcomes, there was an overall improvement in some or all aspects of knowledge and all trainees endorsed PFA training as helpful for increasing their preparedness. PFA training increased trainee’s knowledge and understanding of the appropriate psychosocial responses to traumatic events, how to provide support to individuals facing adversity and the need for self-care in stressful contexts. Improved knowledge acquisition and retention was reported to be sustained at 3-month and 6-month follow-up in a cluster randomised controlled trial compared with the baseline assessment [42].

An improvement in PFA skills (i.e., through application of PFA skills in real-world situations or simulated scenarios) was demonstrated in 12 studies. These skills included early detection of mental health distress, approach communication, relaxation techniques and stress management. Everly et al. assessed the trainees’ skills for early detection in a recognition task (via a videotape recognition exercise) indicating that trainees were able to recognise key clinical markers of acute distress with greater accuracy than the mental health-trained respondents in the comparison condition [48].

Twelve studies measured substantial improvements in attitude and awareness about disaster mental health, post-traumatic stress and mental illness. Trainees in all studies showed an improved attitude towards emergency response engagement, renewed awareness of nonverbal communication, readiness to deliver PFA and a clear focus on safety.

#### 3.5.2. Effect on Behaviour

Seventeen studies reported a positive effect on behavioural outcomes including improving self-efficacy, resilience and capability in terms of being able to manage and respond to stress. The behavioural indexes in a one-year follow-up survey indicated that 83.5% trainees provided PFA to someone experiencing an emotional crisis in a non-disaster context (e.g., trauma caused by violence, serious illness or the death of a loved one) and reported an increased ability to cope [56]. One study using the professional’s quality of life scale evaluated at 3-month and 6-month follow-up did not show a significant difference compared with baseline [42].

#### 3.5.3. Effect on Practice

Real world practice changes were evaluated in three studies, where post-training evaluation was conducted at follow-up with patients (or any individuals who suffered acute distress). These studies purported a positive effect on actual practice [43,56,57]. The only evidence available on how PFA training influenced practice from their recipients assessed the trainees’ clinical practice change at 4–6 weeks follow-up using both quantitative and qualitative data [57]. This study using the therapeutic engagement questionnaire investigated the service users who received the healthcare assistants’ services prior to their PFA training and followed up at two weeks later, indicating a statistical improvement in therapeutic engagement (59.6 in mean scores). In addition, when interviewed, their ward managers, reported having observed their staff using their new PFA skills and noticed a renewed enthusiasm and confidence in delivering therapeutic care practice [57].

Two self-report studies using a follow-up survey found trainees were better at responding in simulations of traumatic situations, for example, expressing empathy, differentiating psychological distress and dysfunction and making referrals or advocating for those in need [43,56].

#### 3.5.4. Effect on Satisfaction

Of six studies reporting on PFA trainee satisfaction, findings suggested that the trainees were considerably satisfied (ranged from 68–80%) with the appropriateness of PFA training for preparing them for their role as aid workers, and with the materials, content and delivery of the course [41,43,45,47,57,62].

#### 3.5.5. Effect on System Changes

In addition to training outcomes, changes at system level were reported. In an effort to assess the scalable PFA capacity-building initiatives in America, system impact was also evaluated at local and state government levels, including: (1) translational impact—improved organisation preparedness, increased awareness of emergency response facilities utilisation and strengthened emergency response network by partner engagement; and (2) policy change—acceptance of lay members into professional volunteer corps and investment support for continuous training [56,63,64]. PFA training delivered in response to an Ebola outbreak showed that it was well accepted by non-specialists, addressed significant gaps of mental health professionals and was considered cost-effective in low-resources settings, and therefore a way to strengthen in-country capacity of emergency response [60].

### 3.6. PFA Training Implementation Barriers and Enablers

Three qualitative studies [39,59,60] and three mixed-method studies [41,46,56] reported PFA training implementation barriers and enablers.

#### 3.6.1. Potential Risks

Potential risks from inappropriately delivering PFA training was revealed from PFA trainees’ practice, such as confusion about role boundaries, inadequate skills development and inconsistent use of PFA guidance, which could potentially harm a person in distress [46,60]. Shortened PFA training led to insufficient understanding of the challenges in applying PFA [39]. PFA providers helping hurricane survivors reported that they often hesitated or over-estimated their roles in responding to survivors’ needs and felt they were unprepared to understand cultural differences [46]. Findings from a 1-day PFA training programme tailored to dealing with an outbreak of Ebola described potential risky practices in relation to the application of Hobfoll’s essential principles [60]. For example, giving a false reassurance to calm a distressed person by promising that everything would be fine which undermined their credibility and was counterproductive. Meanwhile, they also found trainees’ attempt to solve the problem for a distressed person rather than promoting the person’s own coping abilities, which could decrease self-efficacy and ultimately hinder recovery.

The major strengths of PFA is that it is regarded as a low-cost intervention and is sufficiently simple that non-mental health workers can easily acquire PFA knowledge and skills, however, the limited training time, absence of post training supervision and lack opportunity for putting PFA skills into practice could all potentially reduce its efficacy. Horn et al. reported findings from large PFA training initiatives delivered during an Ebola outbreak which suggested that PFA knowledge and skills were not easily acquired for these reasons and should be treated with caution [60].

#### 3.6.2. Barriers and Enablers Factors to the PFA Training

This review identified limited implementation barriers which made it difficult to ensure proactive preparedness and capacity building. Firstly, difficulties of delivering high quality PFA training in emergency situations which had frequently led to delivering just-in time PFA training out of necessity [60].

Secondly, there were other obstacles to delivering high quality PFA training delivery, such as time constraints (e.g., workload made attending training difficult), lack of site-specific training content, limited organisational support due to the misinterpretation of PFA training by funders and confused conception of PFA [59]. Finally, Semlitz also found that lack of qualified PFA trainers and an underestimation of cultural differences in Japan might hinder the sustainability of the training [41].

Conversely, some studies reporting qualitative findings highlighted enabling factors which might improve the PFA training spread. Firstly, contextualisation/adaptation of the PFA training materials to the target trainees for which they were intended would be beneficial [60]. Clarification of the learning objectives with context-tailored content could sustain interest and promote uptake both at individual and organisational levels. In particular, Horn suggested that government standardisation of PFA training and integration in national emergency response structures/systems were important to strengthen organisational capacity [60]. Additionally, short yet ongoing training, using an existing training platform at prescheduled training time, and certain incentives (free training, credits) could be a feasible method of sustaining the training. A mixture of approaches (e.g., online, face-to-face, interactive, role play, real-world application) supplemented with materials in a short-term ongoing training (e.g., conferences, agency meetings, in-service days) was preferred by trainees [59]. Chandra pointed out that online back-up information system with referral resources, mobile PFA tips, lecture slides and evaluation tool with community were welcomed by disaster volunteers [46].

## 4. Discussion

This scoping review of Psychological First Aid training has mapped out the current PFA training application, outcomes and its implementation by reviewing available empirical evidence. The inadequate guidance on how PFA training should be applied and adapted, significant shortcomings of reporting PFA training delivery, limited training evaluation and unclear training outcomes have been highlighted.

### 4.1. Lack Guidance on PFA Training Application and Adaptation

Even though various models existed, three PFA models (WHO-PFA, NCTSN-PFA, RAPID-PFA) were most influential and each targeted different priorities and were developed for use in different populations and settings. For example, the WHO-PFA training manual introduced “Look, Listen, Link” action steps into 20 languages for use in various international settings, whereas training in the Johns Hopkins RAPID PFA model represented proactive preparedness efforts to enhance surge capacity during disaster scenarios in America [25]. As a general observation, there is currently insufficient clear guidance on how to adapt and implement these models for use in different populations and settings.

Firstly, the multitude of different PFA training resources brings complexities in interpretation which may lead to confusion for implementors [10,59]. Given this, it is increasingly common for trainers to integrate different PFA models when attempting to adapt PFA for their specific populations or settings [52,54], and in doing so, the fidelity to the core components of the PFA models remain unclear.

Secondly, PFA is predominately used in Western counties and humanitarian settings. PFA tends to be seen as only as a disaster frontline approach for first responders [10], which could further deter the use by other types of health workers, i.e., non-first responders’ groups.

Thus, despite an emerging consensus for making PFA a mandatory, pre-requisite training for all health workers to ensure frontline support [23], guidance on how to adapt PFA materials and deliver training for use in different settings and populations has not yet been developed. We would recommend that this work is urgent.

### 4.2. Significant Shortcomings in Reporting PFA Training Intervention

This review is the first to provide a detailed profile of current PFA training delivery using the TIDieR checklist and has revealed a striking variability. The significant shortcomings of reporting this educational intervention include a lack of clarification around the theoretical basis of the training, differences in training intensity and limited or no fidelity checks. This in turn leads to challenges for implementors, for example, no ideal length/duration for PFA training leads to varied interpretations, so training intensity can range from 90 min to 6 days. Additionally, in reporting their findings, many authors of the included studies do not provide adequate training content description or, information about how to access this information, which is likely to hamper the replicability of PFA training.

As a consequence of vague descriptions of PFA training content and delivery, it further highlights the difficulties of evaluating fidelity to the original PFA model. Thus, better reporting of PFA training interventions is essential to enable replication, implementation and also facilitate understanding the effectiveness, as there is still a lack of evidence on the use of PFA as a therapeutic intervention [66].

### 4.3. Unintended Potential Harms from Inappropriate Training Delivery

The unintended potential harms from inappropriate PFA training delivery identified by this review need to be considered by all stakeholders in order to support future uptake. These potential harms arise from trainees’ confusion about role boundaries in practicing PFA (e.g., how to assess and decide whether to refer to mental health services?). It is conceivable that inconsistences in PFA guidance could damage the widely held perception of PFA being a “good thing”. Similarly, a lack of training evaluations, and shortened and non-observable PFA training courses have reinforced the perception of PFA as easy to learn and cheap to deliver [60], and diluted versions of PFA training courses are increasingly common.

Baron argues that psychosocial training courses need to provide trainees with adequate opportunity to rehearse and practice [67], for instance, providing trainees with the opportunities to receive post training PFA supervision might be necessary. This review raises questions for future research, on whether insufficient PFA training without supervision leads to adequate skill development. Our review suggest action is needed to address these challenges in order to retain PFA as a humane, do no harm approach to support distressed ones, which is similar to the conclusions of a previous five-year retrospective report [10].

### 4.4. Limited Efforts in Evaluation of PFA Training

There is reasonable evidence in the effectiveness of the PFA training that it significantly improves both trainees’ knowledge of what an appropriate psychosocial response is and PFA skills to provide support to individuals exposed to acute adversity, thereby enhancing self-efficacy and promoting resilience. However, despite the popularity of the term “psychological first aid” in published literature, a limited number of robust evaluations have been taken. As our broad search returned near 3000 articles, only 23 studies met the inclusion criteria of having the training course having been empirically evaluated. Among these, over two third of PFA training evaluations were conducted in the high-income countries and mostly among the official disaster first responders such as NGO workers. This is due to their advanced systems receiving long-term funding, and a comprehensive capacity-building initiative through guidelines and strategic policy mechanisms [68]. Few evaluations have been undertaken in regions that experience frequent emergencies, such as Asian countries or in populations of healthcare sector workers who are the backbone workforce that are called out to respond to a range of public health emergencies and disaster situations.

We also found that there was a huge variability of research designs and outcome measures used to evaluate PFA training courses. Unfortunately, studies with larger samples and a longitudinal design or randomised trial designs were scarce. In particular, given that numerous online learning platform offering PFA training now exist, little is known about the feasibility, effects and costs of these online learning platforms comparing to other types of training delivery formats. This may explain why such preparedness and prevention efforts are not always prioritised in health care approaches by funders and policymaking [69]. Given the lack of formal consideration for evaluation, it may be helpful to develop evaluation protocols to inform local stakeholders, identify PFA training focal points and encourage collaboration through joint planning, monitoring and evaluating.

### 4.5. Less Defined Clear Outcomes for Evaluating the PFA Training

This review has classified the outcome measures for PFA training evaluation which include not only domains of wellbeing such as quality of life, resilience, but the broader range of knowledge, attitudinal and behavioral outcomes. This could serve as a basis for future more targeted selection of outcome measures to identify the anticipated and desired benefits. Secondly, this review shows that evaluators favored measures of knowledge acquisition and preparedness over skill development of the trainees. Importantly, researchers have evaluated less real-world application of PFA skills and knowledge via the measurement of service users’ outcomes. There is a need for future robust research studies to evaluate the application of PFA knowledge and skills in real world settings as part of achieving population-level crisis mental health preparedness.

### 4.6. Implications for Research and Policy

The following inter-linked future research questions arise from this review:

What are the ‘core’ and ‘adaptable periphery’ components of PFA?Is PFA training sufficient to avoid unintended harms and should trainees be given supervision whilst they learn to put training into practice?Is PFA training effective and at what levels and which population?

The discussion of this review on solutions is in line with global discussions about the need to task-shift at least a portion of mental health care to non-specialists [70,71], and the integration of Mental Health and Psychosocial Support (MHPSS) into disaster risk reduction efforts [72]. Especially as mental health preparedness plays a crucial role in responding to public health emergencies across countries, better dissemination and advocacy to policy makers about evidence based PFA training intervention are essential pillars of prevention efforts.

### 4.7. Strengths and Limitations

There are several strengths to this review. Firstly, the review has identified the range of settings and populations in which PFA has been applied. Secondly, the use of TIDieR checklist enabled us to compare and contrast how PFA training for mental health preparedness has been designed and delivered. Thirdly, the review has classified and summarised the outcome measures used to evaluate PFA. Lastly, the review has identified barriers and enablers to implementing PFA, including potential unintended harms.

Several limitations are also apparent. Firstly, the TIDieR checklist highlighted poor adherence to CONSORT guidelines for accurately reporting PFA training intervention in some studies and attempts to contact several authors for more information went unanswered. Secondly, only standard indexed sources of published literature were searched, so the search strategy may not have identified all relevant studies and other “unevaluated” PFA training programmes may have been omitted.

## 5. Conclusions

Research evidence of reasonable quality demonstrates that PFA training significantly improves knowledge of appropriate psychosocial response and PFA skills in supporting people in acute distress, thereby enhancing self-efficacy and promoting resilience. The lack of guidance on how PFA training is applied and adapted, significant shortcoming of reporting PFA training delivery, lack of robust PFA training evaluation and clarity around outcomes has been highlighted. Whilst behavioral, knowledge and system impact of the PFA training are promising, methodologically stronger evaluations which include systematic training adaptation and selection of sensitive outcome measures is needed to strengthen future implementation of PFA training and thereby enhance population preparedness for future emergencies.

## Figures and Tables

**Figure 1 ijerph-18-04594-f001:**
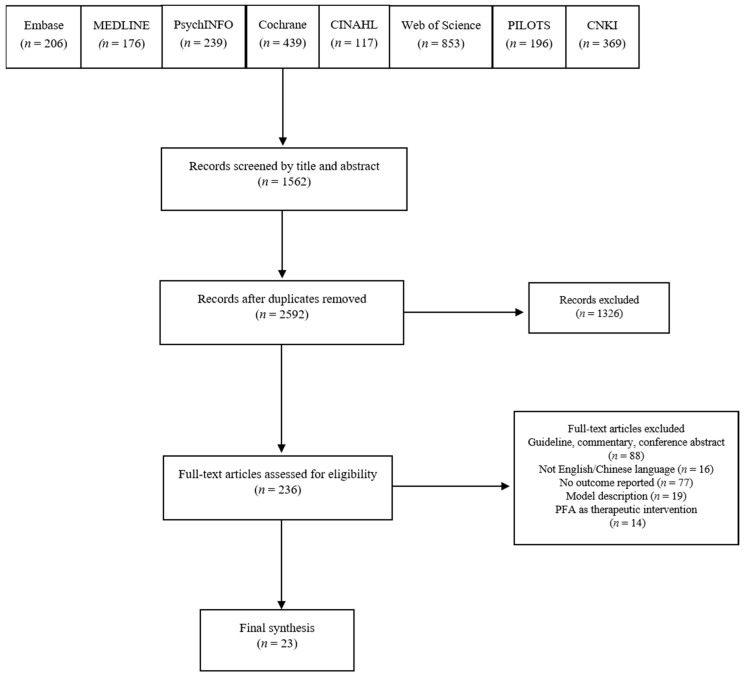
PRISMA flowchart.

**Table 1 ijerph-18-04594-t001:** Inclusion and Exclusion criteria.

	Inclusion Criteria	Exclusion Criteria
Population	Any studies that sampled PFA trainees. e.g., disaster responders, health professionals, decision-makers.	None.
Intervention	PFA training models which adhered to the five Hobfoll’s principles of an immediate mass trauma intervention to improve: safety, calming, efficacy, connectedness and hope [13].	None.
Comparison	Studies were not required to have a control or comparison group, due to the exploratory nature of the scoping review.	None.
Outcomes	All outcomes investigated by the training courses, delivered at all intervention levels: individuals, organisation and system.	None.
Studies	All study designs evaluating PFA training, e.g., qualitative, quantitative and mixed-method studies.	Non-English and Chinese language sources.

**Table 2 ijerph-18-04594-t002:** Comparison of different PFA model.

	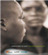 ⟪Psychological First Aid Guide for Field workers⟫	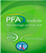 ⟪Psychological First Aid: Field Operations Guide⟫	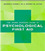 ⟪The Johns Hopkins Guide to Psychological First Aid⟫	Immediate Cognitive-Functional Psychological First Aid (ICF-PFA): The SIX-Cs PFA Model
**Developer**	**World Health Organization** **War Trauma Foundation** **World Vision International**	**National Child Traumatic Stress Network** **National Centre for PTSD**	**Johns Hopkins University**	/
**Intended purpose**	**Humanitarian aid in low- and middle-income countries**	**Common disaster mental health response and research**	**Mental health preparedness and community resilience initiatives**	**Nonprofessional community,** **Professionals and first responders**
**Approach**	Look: Check for safetyCheck for serious distressed peopleApproach people who may need help	Contact and Engagement	R-Establishing Rapport and Reflective Listening	Cognitive verbal communication
**Listen**	Listen: Listen to people, and help them to feel calm	Safety and ComfortStabilisation	A-Assessment/Listening to the story	Listening the answers to the above cognitive questions
**Assessment**	Assessing needs and concerns	Information Gathering on Current Needs and concerns	P-Psychological Triage/Prioritisation	Set priorities and make effective decisions
**Support**	LinkHelp people address basic needs and access servicesHelp people cope with problemsGive informationProtecting people from further harm	Practical AssistanceConnection with Social SupportsInformation on Coping	I-Intervention Tactics to Stabilise and Mitigate Acute Distress	Challenge & control:Ask to choose from simple available actions/options Commitment for the person’s safety
**Referral**	Prepare: Developing a list of available services and support in areasLink: Connect people with loved ones and social support	Linkage with Collaborative Services	D-Disposition and Facilitating Access to Continued Care	Continuity: chronologicalSynchronisation of the event: underline the ending of the event
**Self-care**	Caring for yourself and your colleagues	Provider care	Self-care	/
**Available resources**	Psychological first aid: guide for field workersFacilitation Manual: Psychological First Aid during Ebola Virus Disease OutbreaksPsychological First Aid for Ebola Virus Disease OutbreakPsychological first aid: facilitator’s manual for orienting field workers **31 languages**	Psychological First Aid (PFA) Field Operations Guide for Community Religious ProfessionalsPsychological First Aid (PFA) Field Medical Reserve Corps Field Operations GuidePFA: Parent Tips for Helping School-Age Children after DisastersPsychological First Aid (PFA) for Families Experiencing Homelessness **6 languages**	**Book**The Johns Hopkins guide to psychological first aid. **Online course: Coursera**	/
Evaluation evidence support	Earthquake, Conflict, Triple Disasters, Refugee, Ebola, Acute wards	Hurricane, Flood, Fire, Typhoon, Refugee, Crime, Nursing Home staff	Emergency response, Mental health preparedness capacity building, College student	War crisis, High school student

**Table 3 ijerph-18-04594-t003:** Characteristics of included studies (*n* = 23).

First Author/Year/Country	Aim/s	Sample/Context	Methodology	Outcomes	Key Findings
Allen et al. [43] (2010) USA	To examine the perceptions of the providers after PFA training	Disaster workers (*n* = 50) Hurricane	Cross-sectional post-test study	Survey: scale 1 on perceptions of PFA utilisation and satisfaction scale 2 on perceptions of PFA core actions	Perceptions of using PFA (*p* < 0.01): confidence increased in working with adults and children (0.66); perceived appropriateness of PFA (0.72); overall satisfaction with PFA (0.74); Most helpful to survivors: safety and comfort (72%); practical assistance (44%) and contact & engagement (42%); Most helpful to providers: contact &engagement (64%), safety & comfort (42%) and practical assistant (42%).
Akoury et al. [44] (2015) Lebanon	To examine the efficacy of a national PFA training program to enhance the readiness of mental health field workers in Syrian refugee response	Government, NGO workers (*n* = 60)	Pre- and post-control study	Survey: evaluation form A on knowledge and skills evaluation form B on readiness of the trainees	Significantly increased in knowledge about PFA principles and actions (*p* < 0.001); significantly increase readiness, specifically on the components related to the principles and techniques of PFA (*p* < 0.001).
Brown et al. [45] (2009) USA	To evaluate the feasibility of a modified PFA training intervention among nursing home staff.	Nursing home staff (*n* = 22)	Mixed methods study	Survey: programme evaluation Qualitative discussion	Survey findings: PFA guide was successfully tailored to fit the needs of nursing home residents; all participants rated with “strongly agree” that it was well organised, easily understood, readable and comprehensive; 55% respondents rated willingness to be the PFA trainer and 77% planned to train others; Qualitative findings: increased knowledge about disasters; recommend to colleagues; increase the ability to provide PFA
Chandra et al. [46] (2014) USA	To assess the feasibility and impact on knowledge, attitudes; To explore perceived facilitators and barriers to improve PFA training	Medical response corps volunteer (*n* = 76)	Mixed methods study	Survey: surveys on knowledge, attitudes, and reported practices Qualitative: focus group to discuss the facilitators and barriers	Survey findings: barriers to provide psychosocial support: lack of training (68%), access issues (52%), concerns about comfort or stigma; perceived confidence and capability in using PFA increases from 71% to 90% (*p* < 0.01); no significant increase in PFA-related knowledge; Qualitative findings: Increased intention to use PFA; Improved confidence and competency in addressing post disaster needs.
Cheung Y.L. [47] (2014) China	To investigate the effectiveness of pre-disaster training of PFA among first responders in enhancing PFA provision to survivors of disasters	First responders from Auxiliary Medical Service (*n* = 624)	Randomised controlled trial	Survey: Primary outcome: actual helping behaviour; self-efficacy; knowledge Secondary outcome: psychological distress; coping and social support; life satisfaction and resilience	Significant improved in (a) knowledge on PFA and disaster mental health (*p* < 0.05), (b) self-efficacy (*p* < 0.05) and (c) resilience Unchanged remain (a) level of psychological distress, (b) coping and social support throughout T1,2,3
Everly et al. [48] (2012) USA	To evaluate the attitudes, knowledge and behaviour towards the content. validation of the PFA training	Community citizens (*n* = 252) Mental health preparedness	Pre-/post-test study	Survey: Attitudes; knowledge; behaviours: ability to recognise clinical markers	Significantly improved in self-perceptions of both confidence and preparedness in the application of PFA; knowledge related to the application of immediate mental health interventions; self-perceptions of personal resilience (*p* < 0.001); Improved ability to recognise clinical markers in the field (*p* = 0.027).
Everly et al. [49] (2014) USA	To determine whether relevant knowledge, skills, and attitudes are imparted to trainees to support the effectiveness of PFA training	Public health personnel (*n* = 1500) Mental health preparedness	Pre-/post-test design study	Survey: Knowledge; skills; attitudes	Significant pre-/post improved in (a) knowledge supportive of PFA delivery, (b) perceived self-efficacy to apply PFA and (c) confidence about being a resilient provider (*p* < 0.001).
Farchi et al. [55] (2018) Israel	To evaluate general self-efficacy, self-efficacy, resilience and perceived stress using the training among students	High school students (*n* = 232)	Pre-/post-test, controlled study	Survey: Self-efficacy Resilience The Perceived Stress Scale (PSS)	Successful in providing and maintaining improved GSE, resilience and reduced levels of perceived stress in the long term; Improvement in actual resilience and reduced perception of stress.
Hambrick et al. [59] (2014) USA	To gain understanding of training needs and preferences; To identify challenges and facilitators towards successful dissemination.	Disaster mental health providers (*n* = 9)	Qualitative study	Qualitative interview: Disaster response and preparedness mental health team and resources Training nuts and bolts Desired training focus Training sustainability	Broad awareness and collaboration of organisations; commitment to disaster mental health; lack of resources (time and funding) to provide training; Desire a mixture of approaches (e.g., online, face-to-face, interactive, role play, discussion of practice) with material for personal review; Preferred short yet ongoing training; just-in-time training; Preferred child-focused training, evidence-based; Enablers: evaluations, certifications, available internal trainers and local provision of the trainings; awareness improvement from providers; clarify the applicability and usefulness of the training and certain incentives would increase training attendance.
Horn et al. [60] (2019) Sierra Leone	To understand whether the training provided enabled non-specialists to incorporate the key principles into their practice.	PFA trainers (*n* = 24) PFA trainees (*n* = 36) Key informants (*n* = 12)	Qualitative study	Qualitative interview: Cultural adaptation of the PFA approach Rolling out PFA training Effectiveness of the PFA training Factors contributing to effectiveness of PFA capacity-building	Qualitative findings: (1) Cultural adaptation Only terms of language and role play were modified based on the PFA manual (Ebola version), training greatly reduced into a 95-min time slot; (2) Rolling out training “Emergency” result in confusion: poor-quality of training (diluted), multiple trainings, short train-of-trainers rarely included training skills; Varied in nature and quality of the refresher training and supervision; (3) Effectiveness of training: responses were less consistent with the guidance, e.g., calming people or promoting hope by reassuring them and making false promises; challenging by helping them to promote self-and collective efficacy and connectedness; (4) Challenges to the effectiveness of PFA capacity-building: difficulties in coordinating training activities; confusion of the purpose and limitations of PFA; unavailable supervision result in diluted PFA delivery.
Kılıç et al. [52] (2019) Turkey	To investigate the impact of PFA training on the perception of disaster preparedness and self-efficacy	Nursing student: Intervention group with PFA training (*n* = 38) Control group with no training (*n* = 38)	Randomised controlled trial	Survey: Disaster preparedness Self-efficacy	Positively affected perception for disaster preparation and perception of general self-efficacy in the intervention group (*p* < 0.01); Positive significant relationship between general self-efficacy and disaster preparedness (*p* < 0.05).
Lewis et al. [61] (2013) Australia	To examine the impact (knowledge, skills and overall understanding of response) of PFA training for both managers and peer supporters in a high-risk organisation	Police force (*n* = 582) High-risk organisation	Pre-/post-test design study	Survey: knowledge about post-trauma event (PTE) and PFA; self-reported knowledge and skill about response; confidence in emergency response	Immediate effects of improving both manager and peer supporter knowledge and skills related to PTE and PFA; Increasing capacity to implement a PFA response through increases in relevant knowledge, skills and confidence.
Lee et al. [53] (2017) Korea	To examine the training effects of a didactic and simulation based PFA program	College student (*n* = 37) Counsellor (*n* = 82)	Quasi-experimental pre-and post-control design	Survey: PFA knowledge Perceived competence in PFA skill Perceived willingness, preparedness and confidence	All improved in both groups, except the counsellors’ perceived willingness to provide psychological response for future disaster showed no significantly change after the training.
Mosley et al. [62] (2007) USA	To determine the PFA training model feasibility, perceived effectiveness and prospective translational impact	Faith members (*n* = 500) Mental health preparedness	Quasi-experimental design	Survey: Feasibility: process evaluation Effectiveness: self-efficacy; overall impression of the programme	Desire of the training in the observation that more than double number of anticipated participants Significantly increased knowledge of disaster mental health concepts; skills (self-efficacy); capabilities of leading disaster preparedness planning efforts.
McCabe et al. [63] (2011) USA	To assess the feasibility and effectiveness of a system based PFA training among lay citizens	Community citizens (*n* = 178) Mental health preparedness	Quasi-experimental design	Survey Feasibility: willingness to participate, readiness, ability Effectiveness: knowledge; skill; attitudes Impact: policy change	Feasibility: practicable training model Effectiveness: 93%–99% individuals agreed or strongly agreed that high levels of perceived self-efficacy, ready, willing and able to mental health response. Policy change to recognise and recruit lay members into professional volunteer corps.
Mosley et al. [58] (2012) USA	To determine the feasibility of a brief PFA training to enhance the perceived self-efficacy among faith leaders	Faith leader (*n* = 72) Mental health preparedness	Quasi-experimental design	Survey: Feasibility: process evaluation Effectiveness: knowledge, perceived self-efficacy; impression of the programme	Especially effective training reported with “Very Good or Excellent”; Significantly enhanced knowledge and skills associated with 3 PFA modules (71.2%–96.4%); positive attitudes about the quality and usefulness of each module; increased their confidence in support future victims of trauma.
McCabe et al. [56] (2014) USA	To evaluate the feasibility and the effectiveness of PFA training among lay citizens	Lay citizens (*n* = N/A) Mental health preparedness	Pre- and post-training, mixed methods study	Survey: Effectiveness: knowledge, skill, attitudes; Impact	Significantly improved knowledge and skills in technical or practical PFA content (*p* < 0.001); Changes in policy of government and faith partners to sustain the training to promote public health preparedness and community resilience.
McCabe et al. [64] (2014) USA	To assess the outcomes of 1-day PFA training workshop among urban and rural communities	Urban and rural communities (*n* = N/A) Mental health preparedness	Pre- and post-training, mixed methods study	Survey: Knowledge, skill, attitudes (KSA); behavioural index; sustainability and model scalability Impact: observations	Significant improved in KSA (*p* < 0.001); significant self-reported improvements in general disaster mental health literacy supporting PFA proficiency (*p* < 0.001); 1-year follow-up measure in real-trauma context: 83.5% provided PFA; 84.9% more confident, better at empathy and differentiating psychological distress and dysfunction. Policy and practice change at local and state level.
Schafer et al. [39] (2010) Haiti	To understand the experience from the pilot PFA use within the Haiti context	NGO staff (*n* = 119) earthquake	Qualitative study	Qualitative interview: the usefulness of the source material; the application of the PFA approach; contextual lessons and the limitations.	Basic principles and simple language make material useful in emergency; Positive changes in communication; increased understanding of the limits and extent of the psychosocial assistance; improved confidence to work with distress individuals; Limited availability for a full day training in emergency; unavailability of challenge sharing in shortened training.
Semlitz et al. [41] (2013) Japan	To examine the utility and effectiveness of a culturally adapted PFA training	NGO staff (*n* = 463) Triple disaster	Mixed methods study	Surveys: training needs questionnaire on knowledge and perceived self-competency. Qualitative discussions: qualitative questions in acquisition of training and specific issues.	The perceived training needs included gaining accurate and up-to-date knowledge about disaster relief; learning how to support in helpful and effective ways; Significantly increased participants’ perceived competency in applying PFA principles in a safe manner; understanding of self-care need (*p* < 0.001); 80% acknowledged the useful of the training; Prefer a single long day training; underestimated cultural differences in Japan; challenges of self-care within specific organisations; insufficient trainers.
Sijbrandij et al. [42] (2020) Sierra Leone	To extend the evidence base concerning PFA by evaluating the effectiveness of a one-day PFA training in the acute aftermath of adversity.	Primary health workers Ebola Intervention group with PFA training (*n* = 206) Control group without training (*n* = 202)	Cluster-randomised controlled trial	Primary outcomes: knowledge about psychosocial support; understanding of skills; Secondary outcomes: confidence; professional attitude; professional quality of life	Improved knowledge and understanding of appropriate psychosocial responses and skills in providing support to individuals exposed to acute adversity at the follow-up (*p* < 0.0001); No significant overall differences over time between PFA and control for professional attitude (*p* = 0.20), confidence (*p* = 0.64) and professional quality of life (*p* = 0.63).
Xenya et al. [57] (2020) UK	To introduce and evaluate first aid training for healthcare assistants	Healthcare assistant (*n* = 16) Acute inpatient wards	Mixed methods, pre-and post-test study	Survey: self-efficacy; therapeutic engagement Qualitative interview: evaluation from service user and ward manager	Survey findings: Significantly increased therapeutic engagement in healthcare assistants and service users; Significantly increased healthcare assistants’ self-efficacy and confidence, “untoward” incidents decreased; Qualitative findings: useful in improving staff confidence and reflection on practice, calmer ward atmosphere; therapeutic engagement with service users and ward culture improvement.
Zhang et al. [54] (2016) China	To explore the influence of PFA training on cognitive appraisal of sudden events in medical students	College students Intervention group with PFA training (*n* = 42) Control group without training (*n* = 53)	Randomised controlled study	Survey: The Stress Appraisal Measure Looming maladaptive style questionnaire	Significant differences in threat, challenge, controllable by self, stressfulness, physical threat (*p* < 0.05); Certain effects on changing cognitive about sudden events.

**Table 4 ijerph-18-04594-t004:** Summary of PFA trainings and outcomes according to the TIDieR checklist (n = 20).

Study	Ix	Outcome Measures				Why	Who				How and Where				When and How Much							Tailoring	How Well?	Total
		Learning	Behavior	Practice	Satisfaction	Theory	From		To		Deliver			Site	Duration			Number of Sessions				Attrition	Adherence	
							pm	nw	p	np	lec-	com	sup	cl	Short ≤1/2 d	Med 1 d	Long >1 d	Low ≤4 se	Med 4–8 se	Long >8 se				
Allen [43]	NCTSN			√	√	⊘	●		●			●				●		−			◉	⊘	⊘	4
Akoury [44]	NCTSN	√	√			⊘		●	●			●		●			●		●		●	⊘	⊘	7
Brown [45]	NCTSN		√		√	⊘		●	●			●		●	●			−			●	⊘	⊘	6
Chandra [46]	NCTSN	X	√			⊘	⊖		●			●		●		●		−			●	⊘	⊘	5
Cheung [47]	NCTSN	√	√		√	⊘	●			●		●				●		●			◉	⊘	⊘	5
Everly [48]	RAPID	√	√			●	●			●		●		●		●		−			◉	⊘	⊘	6
Everly [49]	RAPID	√	√			⊘	●		●			●		●		●		−			◉	⊘	⊘	5
Farchi [55]	ICF-PFA		√			●	⊘			●							●	−			⊖	⊘	⊘	3
Kılıç [52]	Mixed		√			⊘	●			●		●		●			●		●		⊖	⊘	⊘	6
Lee [53]	Mixed	√	√			⊘	●			●		●		●		●		−			●	⊘	⊘	6
Lewis [61]	NCTSN	√	√			⊘	●		●		●				●			−			⊖	⊘	⊘	4
McCabe [63]	RAPID	√	√			⊘	●			●		●		●		●		●			◉	⊘	⊘	6
McCabe [64]	RAPID	√				⊘	⊘			●		●		●			●	●			◉	⊘	⊘	6
McCabe [56]	RAPID	√	√	√		⊘		●		●		●		●		●		●			◉	⊘	⊘	6
Mosley [62]	RAPID	√	√		√	⊘	●			●		●		●		●		●			◉	⊘	⊘	6
Mosley [58]	RAPID	√	√			⊘		●		●	●			●			●	●			◉	⊘	⊘	6
Semlitz [41]	WHO	√	√		√	⊘	●		●		●			●		●		−			●	⊘	⊘	6
Sijbrandij [42]	WHO	√	X	√		⊘	●		●			●	●	●	−			−			●	●	●	7
Xenya [57]	WHO	√	√	√	√	⊘	⊘		●			●		●		●		−			◉	⊘	⊘	4
Zhang [54]	Mixed		√			⊘	●			●		●					●			●	⊖	⊘	⊘	5
Total		√ 14; X 1	√ 17; X 1	√ 4	√ 6	2	12	5	9	11	3	15	1	15	2	11	5	6	2	1	6	1	1	

Abbreviations: TIDieR, template for intervention description and replication; Ix: PFA training intervention; pm: psychologist, psychiatrist, mental health researchers; nw: non-mental health workers (Train-the-trainers); p: health professionals; np, non-health professionals, student/residents/etc; lec-: didactic lecture only; com: combination, e.g., lecture, simulation, role-play etc; sup, supervision; cl: classroom; med: medium; d: day; se: sessions; ● reported; ⊘ not reported; ⊖ unclear; X, no effect; √, effect; −, not applicable; ◉ adjunct to main PFA training intervention.

**Table 5 ijerph-18-04594-t005:** Outcome measures of PFA training.

Outcomes Measures	Learning	Behaviour	Satisfaction	Practice
**Validated tools**				
The General Self-Efficacy scale (GSE)		+[52] +[55] +[57] +[47]		
The General Health Questionnaire (GHQ)		+[47]		
Depression Anxiety Stress Scales (DASS21)		−[47]		
Brief COPE scale		−[47]		
Professional Self-Efficacy (PSE)		+[55]		
The Connor-Davidson Resilience scale (CD-RISC)		+[55] −[47]		
The Perceived Stress Scale (PSS)		+[55]		
Multidimensional Scale of Perceived Social Support (MSPSS)		−[47]		
Professional Quality of Life Scale (ProQOL-5)		−[42]		
Therapeutic Engagement questionnaire (TEQ)				+[57]
The Stress Appraisal Measure		+[54]		
Looming maladaptive style questionnaire		+[54]		
**Self-designed tools**				
Skill, Knowledge and attitude	+[41] +[42] +[44] −[46] +[49]+[53] +[56] +[58] +[61] +[62] +[63] +[64]			
Ability to recognise clinical markers		+[48]		
Perceived competence		+[40] +[48]		
Scales on willingness, preparedness and confidence		+[44] +[45] +[49] +[52] +[53] +[58] +[61]+[62] +[63]	+[62]	+[49] +[56]
Scale on perceptions of PFA utilisation and satisfaction on PFA training			+[43]	
Programme Evaluation			+[45] +[47] +[55] +[57]	+[43]

Numbers in square brackets are cited reference; +, signifies a positive finding; −, signifies no statistically significant finding.

## Data Availability

The data that support the findings of this study are available from the corresponding author, upon reasonable request.

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
