# Peer review of "Psychological First Aid Training: A Scoping Review of Its Application, Outcomes and Implementation"

_ijerph, 2021, doi:10.3390/ijerph18094594_

Round 1

Reviewer 1 Report

General comments:

This is a really valuable piece of work, as I think it will be very useful for many within the healthcare sector. The review was well-conducted. Some comments for strengthening the manuscript are detailed below.

  1. Reference 3 (Greenberg et al, 2020) does not support the claim made on page 1, lines 40-41 - that moral injury and mental health problems experienced by healthcare workers can result in medical errors. While the link between moral injury and medical errors could be hypothesised, the cited reference does not test the link between these factors.
  2. It would be useful to discuss in the introduction whether 'PFA' is different from 'PFA training', or whether the training itself is the 'first aid'. It is not clear from the introduction whether the training is equipping healthcare workers to deliver PFA to community members, or whether the training involves the healthcare workers themselves receiving 'first aid'. I understand that one of the purposes of this review is to seek clarity on the concept of PFA, but it is not clear from the introduction whether the authors consider 'PFA' and 'PFA training' as synonymous or distinct concepts.
  3. Relatedly, I think it would be beneficial to include a sentence in introduction about the fact that there is no one agreed-upon definition of PFA - you have described the 'broad professional consensus' which hints at the lack of clear definition, but I think it would be helpful to explicitly 'spell this out', per se, especially as this issue is raised in section 2.2.
  4. The research question is not very specific - I am not sure how the "current status" is measured. This could be reworded. Also, it seems from section 2.2. in the Method, that an additional aim was to develop a consensus definition of PFA.
  5. On page 4, line 149, it seems as though there is a word missing at the end of the sentence "...as this was not required in line with the scoping review methodology and scarcity of the effectiveness"
  6. I am not clear what the figures within square brackets in Table 5 represent.
  7. Under the heading 3.6, you describe the third theme as enablers and barrers to improving the training, but in section 3.6.3. these enablers and barriers were moreso associated with the delivery of the training itself. I recommend ensuring consistency throughout. I also suggest removing the word 'factors' from the heading of section 3.6.3.
  8. Could you clarify in the Author Contributions which author(s) did the screening, extraction, and analyses?

A final note:

Grammatical issues are present throughout the manuscript - for example, the word "of" should be deleted in the sentence on page 1, lines 36-37, "A priority group of providing early psychosocial response is frontline healthcare workers" and the word "is" is missing from the sentence on page 2, lines 47-48, "However, PFA as a frontline approach to ensure immediated psychosocial care not new". On page 4, line 152, I believe the word "exact" should be "extract". The sentence on lines 464-466 doesn't make sense as there is a mixture of past and present tenses, and the sentence also appears to have words missing. Also - this is a copy-editing issue, but in some instances COVID-19 is capitalised and other times it is not (instead, written "Covid-19").

Sometimes, the language used is not very clear - for example, on page 2, line 66, it is stated that PFA is suited "to expand cadres for widely early psychosocial response" - I am not sure what this means. Across lines 558-559, the sentence "While this review doubts about the potential of PFA as a good common skill for professional helpers, if there is an absence of service users’ evaluating outcomes" is also unclear.

Author Response

Response to Reviewer 1 Comments

Introduction

Point 1: Reference 3 (Greenberg et al, 2020) does not support the claim made on page 1, lines 40-41 - that moral injury and mental health problems experienced by healthcare workers can result in medical errors. While the link between moral injury and medical errors could be hypothesised, the cited reference does not test the link between these factors.

Response 1: Thank you for pointing it out very carefully.

After reviewing the literature, we accept this, as the link between moral injury and medical error does need further investigation. Therefore, we have clarified text throughout:

  • word changed (“can” to “might”),
  • extra reference (reference 4) added; we have stated that “Apart from their pivotal role and responsibility for patient care, they are also at high-risk facing overwhelming stress situation which can lead them to suffer from moral injury and potential mental health problems, which might have negative consequences contributing to poor quality of patient care and attrition in health professionals”.

We have highlighted changes in red to aid visibility. See page 1, line 40-41.

Point 2: It would be useful to discuss in the introduction whether 'PFA' is different from 'PFA training', or whether the training itself is the 'first aid'. It is not clear from the introduction whether the training is equipping healthcare workers to deliver PFA to community members, or whether the training involves the healthcare workers themselves receiving 'first aid'. I understand that one of the purposes of this review is to seek clarity on the concept of PFA, but it is not clear from the introduction whether the authors consider 'PFA' and 'PFA training' as synonymous or distinct concepts.

Response 2: Thank you for highlighting this potential understanding in manuscript. We have now clarified (in para2+3) the difference between the PFA as a therapeutic intervention and the PFA training as an educational intervention.

We do not consider these concepts synonymous and have now added the following clarification:

“PFA training is an educational intervention that aims to equip healthcare workers with the knowledge and skills to deliver the therapeutic intervention of PFA to individuals who require psychological support after having been in disaster situations.” (please see page 2, line 66-69).

Point 3: Relatedly, I think it would be beneficial to include a sentence in introduction about the fact that there is no one agreed-upon definition of PFA - you have described the 'broad professional consensus' which hints at the lack of clear definition, but I think it would be helpful to explicitly 'spell this out', per se, especially as this issue is raised in section 2.2.

Response 3: Thank you. We accept this and have now added:

“Different models exist and there is no uniform definition of PFA across different models, yet the five Hobfoll’s intervention principles established in 2007 has exerted a profound influence on the emerging field of PFA.” on page 2, line 56-59.

Point 4Grammatical issues:

Point 4.1 the word "of" should be deleted in the sentence on page 1, lines 36-37, "A priority group of providing early psychosocial response is frontline healthcare workers".

Response 4.1: Sorry for this, we have now corrected accordingly. Please see page1, line 36-37. “A priority group providing early psychosocial response is frontline healthcare workers, who are at the forefront of supporting acutely distressed individuals.”

Point 4.2 the word "is" is missing from the sentence on page 2, lines 47-48, "However, PFA as a frontline approach to ensure immediated psychosocial care not new".

Response 4.2: We have now amended: “PFA as a frontline approach to ensure immediate psychosocial care is not new”.  Please see page 2, line 48-49.

Point 4.3 Sometimes, the language used is not very clear - for example, on page 2, line 66, it is stated that PFA is suited "to expand cadres for widely early psychosocial response" - I am not sure what this means.

Response 4.3: Thanks for reading through this point, we have rewritten this sentence: “Given this, the PFA training seems well-suited to delivery within the curriculum for non-specialist frontline workers to achieve better preparedness, in order to ensure a broad psychosocial care for distressed individuals and to build a resilient workforce for dealing with emergencies”. Please see page2, line 76-79.

Point 4.4 this is a copy-editing issue, but in some instances COVID-19 is capitalised and other times it is not (instead, written "Covid-19").

Response 4.4: Thanks for carefully pointing it out, we have now amended all instances of this. Please page 2, line 45.

Methodology

Point 5: The research question is not very specific - I am not sure how the "current status" is measured. This could be reworded. Also, it seems from section 2.2. in the Method, that an additional aim was to develop a consensus definition of PFA.

Response 5: Thanks for correcting this word use, we have now reworded using “current application” (see page 3, line 114), as our review mapped out its use, outcomes and implementation issues which could answer this. We did not have an additional aim to develop a consensus definition of PFA.

Results

Point 6.I am not clear what the figures within square brackets in Table 5 represent.

Response 6: The square brackets are the study reference number using the measurement tools. We have added an explanation sentence below in the table 5.

Point 7. Under the heading 3.6, you describe the third theme as enablers and barriers to improving the training, but in section 3.6.3. these enablers and barriers were more so associated with the delivery of the training itself. I recommend ensuring consistency throughout. I also suggest removing the word ‘factors’ from the heading of section 3.6.3.

Response 7: Thanks, we have kept consistency throughout,

  • Changed the heading 3.6 (page 21, line 431) as “factors contributing to the improvement of PFA training”;
  • removed the word “factors” accordingly, “barriers and enablers to the delivery of PFA training”, please see section 3.6.3, page 21 line 470.

Reviewer 2 Report

The exclusion criteria apply to bibliographic sources that have been selected by fulfilling the inclusion criteria, thus it is not correct that the exclusion criteria are the opposite of the inclusion criteria, because it is a condition that is applied after the inclusion criteria has been fulfilled.

My recommendation is to delete or reformulate the exclusion criteria for “intervention” and “studies” on the same grounds.

About table 2

In the table on page 1 of 29, the first column names the evaluated aspect. This term is in the same line as the article information.

In contrast, the aspects "available resources" and "evaluation evidence support" on page 2 of 29 appear as headings in the next line.

If this is the same table, I consider that this change of formatting may be confusing and therefore propose that the terms "a" and "e" appear at the beginning of the corresponding line.

Author Response

Response to Reviewer 2 Comments

Dear reviewer,

IJERPH-1172147
Psychological First Aid training: a scoping review of its application, outcomes and implementation

Thank you for providing us with such valuable feedback and the opportunity to improve our manuscript. Below, we respond to the comments in detail and explain how we have addressed them in our revised manuscript. Should we not have addressed any comments to your satisfaction, we are open to making further changes but would appreciate further guidance. We have provided a WORD version of the manuscript with tracked changes and a clean pdf version of the manuscript without tracked changes. The line numbers given in the author responses refer to the clean pdf version.

Best wishes,

Ling Wang on behalf of all co-authors

Methodology

Point 1: The exclusion criteria apply to bibliographic sources that have been selected by fulfilling the inclusion criteria, thus it is not correct that the exclusion criteria are the opposite of the inclusion criteria, because it is a condition that is applied after the inclusion criteria has been fulfilled.

My recommendation is to delete or reformulate the exclusion criteria for “intervention” and “studies” on the same grounds.

Response 1: Thank you for the suggestion, we have amended the criteria as suggested (now table 1),

  • deleting the exclusion criteria for “intervention” --“primarily focused on other related concepts (e.g., psychological therapeutic interventions following a crisis, psychoeducation)” to “None.”
  • deleting the exclusion criteria for “studies” --“editorial, opinion, protocols, conference abstracts, or conceptual analysis, letters, and guidelines” to “None.”

Point 2: In the table on page 1 of 29, the first column names the evaluated aspect. This term is in the same line as the article information. In contrast, the aspects "available resources" and "evaluation evidence support" on page 2 of 29 appear as headings in the next line. If this is the same table, I consider that this change of formatting may be confusing and therefore propose that the terms "a" and "e" appear at the beginning of the corresponding line.

Response 2: Thanks for pointing out this. We have now changed our formatting as suggested, in the table 2.

Reviewer 3 Report

This is a useful and timely paper.  PFA is already widely used and becoming more so due to the COVID-19 pandemic.  It can be seen as a ‘quick fix’ by governments and agencies, so this paper exploring the effectiveness of PFA training has the potential to be a corrective to that.

The foundations of the paper are sound – the extraction and analysis of relevant research papers seems to have been conducted systematically.  However, there are some significant errors in the analysis and reporting which would need to be corrected before the paper could be considered for publication.

Introduction

The authors have drawn on a wide range of literature here, and the first paragraph sets the scene well.  However, the definition of PFA at the beginning of the second para is unclear – it implies it is primarily a self-care approach, whereas it is, in fact, a way of supporting others in distress.  A clear explanation of PFA at this point in the paper is essential.

It is also important to highlight in the introduction that there are different models of PFA – as becomes clear later in the paper.  A chronological explanation of the development of the different models here would provide the context necessary to understand subsequent sections of the paper.  This chronological approach is important because the systematic review conducted by WHO identified crucial features which were integrated into the 2011 PFA guidance, but had not necessarily been part of earlier versions of PFA.  This means that any critique of the approach which was published before 2011 was referring to a different model, and this should be made clear by the authors.

It is primarily the second paragraph of the introduction which requires revision.  Para’s 3 and 4 are clear, and the rationale for and aims of the study being reported are also clear.

Materials and method

This section is clearly described.  My only query is around describing PFA training as an ‘educational intervention’.  This implies it is a theoretical training, whereas in fact it is a skills training. 

Results

Section 3.2 is confusing and misleading in places.  The citations here are wrong – reference [12] should be [15], and [13] should be [16].  The first two sentences are very difficult to understand, and should be rewritten.  The point being made in this section, which is an important one, is that there are differences between the PFA models used in the training.  Table 2 is extremely helpful - but the authors have not analysed or reported on the three PFA models correctly, and should go back to each model and revise this table so that it accurately reflects their content.  For example, the WHO PFA Guide for Fieldworkers includes a substantial section on self-care, with a whole chapter on ‘caring for yourself and your colleagues’.  The ‘Link’ action principle includes making referrals to enable people to access other services, and the ‘Prepare’ stage includes developing a list of available services and supports in the area, in order to facilitate referrals.  Table 2 states that the WHO version of PFA includes no content on self-care or on referrals.  These fundamental errors need to be corrected.  As a more minor point, this guide was developed by WHO, World Vision and War Trauma Foundation – not World Trauma Foundation, as stated on line 184.

Section 3.3 is good, and section 3.4 is mainly clear and provides very helpful information.  Given the current focus on providing remote PFA training in the COVID-19 situation, it would be helpful to know whether any of the training was delivered online.  Section 3.4.4 states that face-to-face training was most common – but does not mention what the alternative modes of delivery were, and how many cases were not face-to-face.

Table 4 is very helpful.  Three of the studies are missing from this table; it would be good to mention the reasons for this.

Section 3.5.3 is very important, since effect on practice is the key outcome variable of interest, and is the area where there is the least evidence up to now.  The authors should make clear how effect on practice was assessed by the four studies they report on in this section.  In at least one case the study used hypothetical scenarios rather than use of the PFA skills in a real-life situation.

Section 3.6.2 is very important, but the risks are not explained clearly and the findings of at least one of the studies are misrepresented here.  The second sentence is inaccurate – debriefing and grieving are not PFA skills.  Similarly, lines 408-411 do not make sense and do not represent the findings of this study.

Section 3.6.3 is also very important, but since it is based on the findings of only three studies, the conclusions should be expressed much more tentatively.  The fact is that we know little about barriers and enablers to PFA training, but these three studies highlight some potentially relevant issues.

Discussion

There are many good points in the Discussion, but they would come through more clearly if the section was reduced.  It is currently too long, and there is unnecessary repetition.  Section 4.1 is difficult to understand – the content is useful, but the way the sentences are constructed obscures the meaning.  But in general, the Discussion makes some excellent arguments.

Style

The grammar and wording within the paper reads awkwardly in places, and is misleading in others.  The paper would need to be checked for grammatical accuracy before publication.

Summary

This paper has the potential to be extremely useful to people working in the field of mental health and psychosocial support.  There are some significant revisions which are required before the paper could be considered for publication, but I hope the authors will be able to do so – I would very much like to see this scoping review published, to add to the body of evidence around PFA.

Author Response

Response to Reviewer 3 Comments

Dear reviewer,

IJERPH-1172147
Psychological First Aid training: a scoping review of its application, outcomes and implementation

Thank you for providing us with such valuable feedback and the opportunity to improve our manuscript. Below, we respond to the comments in detail and explain how we have addressed them in our revised manuscript. Should we not have addressed any comments to your satisfaction, we are open to making further changes but would appreciate further guidance. We have provided a WORD version of the manuscript with tracked changes and a clean pdf version of the manuscript without tracked changes. The line numbers given in the author responses refer to the clean pdf version.

Best wishes,

Ling Wang on behalf of all co-authors

Introduction

Point 1: The authors have drawn on a wide range of literature here, and the first paragraph sets the scene well.  However, the definition of PFA at the beginning of the second para is unclear – it implies it is primarily a self-care approach, whereas it is, in fact, a way of supporting others in distress.  A clear explanation of PFA at this point in the paper is essential.

Response 1: We have now further clarified the definition of PFA at the beginning of the second paragraph—added “support people in need” as emphasizing PFA as a supportive skillset. Please see page2, line 47.

Point 2: It is also important to highlight in the introduction that there are different models of PFA – as becomes clear later in the paper.  A chronological explanation of the development of the different models here would provide the context necessary to understand subsequent sections of the paper.  This chronological approach is important because the systematic review conducted by WHO identified crucial features which were integrated into the 2011 PFA guidance but had not necessarily been part of earlier versions of PFA.  This means that any critique of the approach which was published before 2011 was referring to a different model, and this should be made clear by the authors. It is primarily the second paragraph of the introduction which requires revision.  Para’s 3 and 4 are clear, and the rationale for and aims of the study being reported are also clear.

Response 2: Thank you, we have added introductory sentences to provide the context as suggested. We have:

1) Added the point “different PFA models exist” on page2, line 56-57.

2) Introduced the five Hobfoll’s intervention principle established in 2007 which has been considered laying a foundation for the subsequent PFA models adoption. Then, we took two exemplar PFA models afterwards to justify the impact of this established principle.

We therefore expanded our explanation of PFA development. Please see page 2, line 54-60.

Methodology

Point 3: This section is clearly described.  My only query is around describing PFA training as an ‘educational intervention’.  This implies it is a theoretical training, whereas in fact it is a skills training. 

Response 3: Thank you. We have thought about it substantially.

As we sought to snapshot the empirical evidence of PFA training, the terminology of education intervention we adopted was implicated by a systematic review published in Lancet psychiatry (REF: mental health training for health workers in Africa: a systematic review). This review has introduced this education interventions as didactic and interactive, assessed knowledge, skills, practice and patient outcomes, hence, we adopt it given we did not find a better term.

Also, we stated it in the “Introduction” para ““PFA training is an educational intervention that aims to equip healthcare workers with the knowledge and skills to deliver the therapeutic intervention of PFA to individuals who require psychological support after having been in disaster situations.” (please see page 2, line 66-69).

Results

Point 4: Section 3.2 is confusing and misleading in places.  The citations here are wrong – reference [12] should be [15], and [13] should be [16]. 

Response 4: Apologies, as we double-checked all the citation and numbers according to the comments, this mistake has been amended. Please see page 5, line 185-188 “namely Psychological First Aid Guide for Field Workers [14], Psychological First Aid: Field Operations Guide [20], The Johns Hopkins Guide to Psychological First Aid [15]”.

Point 5: The first two sentences are very difficult to understand, and should be rewritten.  The point being made in this section, which is an important one, is that there are differences between the PFA models used in the training. 

Response 5: Thank for the advice. We have now rewritten and hope this is clearer. Please see page 5, line 185-191.

Point 6: Table 2 is extremely helpful - but the authors have not analysed or reported on the three PFA models correctly, and should go back to each model and revise this table so that it accurately reflects their content.  For example, the WHO PFA Guide for Fieldworkers includes a substantial section on self-care, with a whole chapter on ‘caring for yourself and your colleagues’.  The ‘Link’ action principle includes making referrals to enable people to access other services, and the ‘Prepare’ stage includes developing a list of available services and supports in the area, in order to facilitate referrals.  Table 2 states that the WHO version of PFA includes no content on self-care or on referrals.  These fundamental errors need to be corrected. 

Response 6: Thank for positive comments and apologies for the error.

Previously, we noticed substantial manuals for each PFA models, therefore, we did just compare the core actions based on the manuals. However, we have realized the difficulty of making a distinct comparison across each model. Thus, we have now changed the content presented in the table 2 by adding the content from each manual. Please see new table 2 on page 6.

Point 7: As a more minor point, this guide was developed by WHO, World Vision and War Trauma Foundation – not World Trauma Foundation, as stated on line 184.

Response 7: Thank you for correcting, we have amended accordingly on page 8, line 199: “This model was jointly developed by the World Health Organisation, World Vision International and War Trauma Foundation”.

Point 8: Section 3.3 is good, and section 3.4 is mainly clear and provides very helpful information.  Given the current focus on providing remote PFA training in the COVID-19 situation, it would be helpful to know whether any of the training was delivered online. 

Response 8: Thank you.

Since our examination of the current PFA training studies, there’s no evidence to inform any information of online training. As requested, we have emphasised this point both in:

  • this section 3.4 (page 15, line 304-305) “Current training evaluation studies had all been done using face-to-face sessions delivered in the workplace”
  • Discussion 4.4 line page 23, 576. “In particular, given that numerous online learning platform offering PFA training exist, little is known about such the feasibility, effects and costs compared to other training delivery formats.”

Point 9: Section 3.4.4 states that face-to-face training was most common – but does not mention what the alternative modes of delivery were, and how many cases were not face-to-face.

Response 9: Thank you. We have reviewed all included studies and confirm that they all provided face-to-face training. We have added the following text:

“all evaluation studies have delivered PFA training sessions face-to-face s”, please see section 3.4.4., see line 304-305.

Point 10: Table 4 is very helpful.  Three of the studies are missing from this table; it would be good to mention the reasons for this.

Response 10: Thank you, we have explained this further to make our point clearer. This is because the three excluded studies are qualitative studies which do not report details of the training. We have added this reason in page 15, line 276-278: “This review excluded the three qualitative studies as they failed to report any details of the PFA training.”

Point 11: Section 3.5.3 is very important, since effect on practice is the key outcome variable of interest, and is the area where there is the least evidence up to now.  The authors should make clear how effect on practice was assessed by the four studies they report on in this section.  In at least one case the study used hypothetical scenarios rather than use of the PFA skills in a real-life situation.

Response 11: We have now made the effect on practice clearer in page 18, line 409-411. “Given the real-life context of their PFA application, there were three studies reporting the follow-up result on training recipients’ practice [43,56,57].”

Also, thanks for pointing out the misleading information from the Sijbrandij et al’s RCT study, we have now deleted it.

Point 12: Section 3.6.2 is very important, but the risks are not explained clearly and the findings of at least one of the studies are misrepresented here.  The second sentence is inaccurate – debriefing and grieving are not PFA skills.  Similarly, lines 408-411 do not make sense and do not represent the findings of this study.

Response 12: Thanks for correcting this misleading sentence.

  • These have been reviewed back to the literature and now amended substantially in section 3.6.2 (please see page21, line 448-468):

 “Potential risks from inappropriate PFA training delivery were revealed, for example that PFA trainees’ practice including confusion about their role boundaries, inadequate skills developed and that less consistent with the guidance could result in potential harms to a person in distress [46,64]. Shortened PFA training courses resulted in participants having difficulties understanding how to apply their PFA training in real world settings [39].PFA providers helping hurricane survivors reported that they often hesitated or over-estimated their roles in responding to survivors’ needs, feeling unprepared to understand cultural differences [46]. Moreover, in an attempt to scale up provision and deliver low-cost PFA training, a 1-day PFA training course was delivered during Ebola outbreak and resulted in some risky practices in relation to Hobfoll’s essential principles [64]. For example, a false reassurance to calm a distressed person by promising that everything would be fine undermined the credibility and was counterproductive. Meanwhile, they also found trainees’ attempted to solve the problem for a distressed person rather than promoting the person’s sense of coping abilities, which could decrease self-efficacy and ultimately hinder recovery.

    Generally, a major strength of PFA is that it is a low-cost, therapeutic intervention that is easily acquired by non-professionals, however, the implementation issues such as limited training time, absence of post training supervision and lack opportunity for putting PFA skills into practice could have an impact on the PFA effectiveness. For instance, Horn et al. reported detailed information from large PFA training initiatives during the Ebola outbreak which suggested that PFA was not easily acquired framework [64]”.

Point 13: Section 3.6.3 is also very important, but since it is based on the findings of only three studies, the conclusions should be expressed much more tentatively.  The fact is that we know little about barriers and enablers to PFA training, but these three studies highlight some potentially relevant issues.

Response 13: Thanks, we have now expressed this more tentatively on page 21, line 471-472: “As limited evaluation of PFA training existed, little were known about barriers and enablers, but these three studies highlighted some potentially relevant issues.”

Discussion

Point 14: There are many good points in the Discussion, but they would come through more clearly if the section was reduced.  It is currently too long, and there is unnecessary repetition.  Section 4.1 is difficult to understand – the content is useful, but the way the sentences are constructed obscures the meaning.  But in general, the Discussion makes some excellent arguments.

Response 14: Thanks for your positive comments in this section.

We have cut down the length and removed the repetition to make it clearer, please see new discussion section on page 22-24.

In section 4.1, we’ve changed the content. please see page 22 section 4.1.

Point 15: Style

The grammar and wording within the paper reads awkwardly in places, and is misleading in others.  The paper would need to be checked for grammatical accuracy before publication.

Response 15: We have now checked and corrected the grammatical mistakes accordingly.

Point 16: Summary

This paper has the potential to be extremely useful to people working in the field of mental health and psychosocial support.  There are some significant revisions which are required before the paper could be considered for publication, but I hope the authors will be able to do so – I would very much like to see this scoping review published, to add to the body of evidence around PFA.

Response 16: Thank you so much for the positive and encouraging comments on our work. We have amended the manuscript substantially based on your valuable advice.

Round 2

Reviewer 3 Report

Thankyou for the opportunity to review the revised version of this paper.  My feedback on the revisions made is below:

Point 1: I’m happy with this revision.

Point 2: The revised version of the introduction is much better than the first one I reviewed.  The authors have satisfactorily addressed the issues I raised.

Point 3: I still have a different opinion to the authors on this issue.  The term ‘educational intervention’ implies to me a theoretical, knowledge based training – but I’ll leave it to the journal editors to decide whether to pursue this revision.

Point 4-10: I’m satisfied with the revisions the authors have made to address these issues.

Point 11:  This remains rather unclear to me.  The authors have focused on 3 studies which followed up on actual practice in an emergency settings, according to the first sentence, but go on to say that two of these studies used self-report to assess changes in practice – which is very different to actual practice.  For clarity, I suggest that the authors differentiate more substantially between the two self-report studies and the single study which evaluated actual change in practice.  Given that this single study is the only evidence available on how PFA training influences practice, it would be useful to explain more about what changes in ‘therapeutic engagement’ were observed (line 421) and which ‘new skills’ were used (line 422).  It would be better to focus on this, rather than report in detail self-reported improvements in empathy etc (lines 412-417).

Point 12: This section (3.6.2) is much improved.  However, inaccurate use of English language and grammar obscures some of the points being made.  This is also the case in other areas of the paper.  I suggest the whole paper is reviewed for its use of English, and improved to enhance clarity.

Points 13-14:  I’m satisfied with the revisions the authors have made to address these issues.

Point 15: As noted, the important points made by the authors would come through more clearly if the English was improved.

I appreciate the efforts the authors have made to address the issues I raised previously.  With a few more small revisions, this paper has the potential to make a useful contribution to the field.

Author Response

Thanks so much again for providing us such valuable points to inform our work. 

Response to Reviewers’ Comments

 IJERPH-1172147
Psychological First Aid training: a scoping review of its application, outcomes and implementation

Thank you for providing us with such valuable feedback and the opportunity to improve our manuscript. Below, we respond to the reviewers’ comments in detail and explain how we have addressed them in our revised manuscript. Should we not have addressed any comments to your satisfaction, we are open to making further changes but would appreciate further guidance. We have provided a WORD version of the manuscript with tracked changes and a clean pdf version of the manuscript without tracked changes. The line numbers given in the author responses refer to the clean pdf version.

Best wishes,

Ling Wang on behalf of all co-authors

Methodology

Point 3: This section is clearly described.  My only query is around describing PFA training as an ‘educational intervention’.  This implies it is a theoretical training, whereas in fact it is a skills training. 

Response 3: Thank you for considering this terminology deeply.

As we discussed based on this thought and from current evidence of PFA Training, it does have a theoretical, knowledge-based component so it's reasonable to call it an 'educational intervention'. Also, we do welcome better terminology by reviewers and editors to inform our future work. Thanks for opening such valuable discussion.

Point 11: Section 3.5.3 is very important, since effect on practice is the key outcome variable of interest, and is the area where there is the least evidence up to now.  The authors should make clear how effect on practice was assessed by the four studies they report on in this section.  In at least one case the study used hypothetical scenarios rather than use of the PFA skills in a real-life situation.

This remains rather unclear to me.  The authors have focused on 3 studies which followed up on actual practice in an emergency setting, according to the first sentence, but go on to say that two of these studies used self-report to assess changes in practice – which is very different to actual practice.  For clarity, I suggest that the authors differentiate more substantially between the two self-report studies and the single study which evaluated actual change in practice.  Given that this single study is the only evidence available on how PFA training influences practice, it would be useful to explain more about what changes in ‘therapeutic engagement’ were observed (line 421) and which ‘new skills’ were used (line 422).  It would be better to focus on this, rather than report in detail self-reported improvements in empathy etc (lines 412-417).

Response 11: Thanks so much for your contribution to our work which led us to consider this point much clearer.

We now revised this section according to your guidance. Please see page 19 line 449-465 and the following:

 “Real world practice changes were evaluated in three studies, where post-training evaluation was conducted at follow-up with patients (or any individuals who suffered acute distress). These studies purported a positive effect on actual practice [43,56,57]. The only evidence available on how PFA training influenced practice from their recipients assessed the trainees’ clinical practice change at 4-6 weeks follow-up using both quantitative and qualitative data [57]. This study using the therapeutic engagement questionnaire investigated the service users who received the healthcare assistants’ services prior to their PFA training and followed up at two weeks later, indicating a statistical improvement in therapeutic engagement (59.6 in mean scores). In addition, when interviewed, their ward managers, reported having observed their staff using their new PFA skills and noticed a renewed enthusiasm and confidence in delivering therapeutic care practice [57].

   Two self-report studies using a follow-up survey found trainees were better at responding in simulations of traumatic situations, for example, expressing empathy, differentiating psychological distress and dysfunction, and making referrals or advocating for those in need [43,56].

Point 12: Section 3.6.2 is very important, but the risks are not explained clearly and the findings of at least one of the studies are misrepresented here.  The second sentence is inaccurate – debriefing and grieving are not PFA skills.  Similarly, lines 408-411 do not make sense and do not represent the findings of this study.

This section (3.6.2) is much improved. However, inaccurate use of English language and grammar obscures some of the points being made.  This is also the case in other areas of the paper.  I suggest the whole paper is reviewed for its use of English, and improved to enhance clarity.

Response 12: Thanks for correcting this again.

   We  made another round proof reading by our British authors and have substantially improved as much as possible in section 3.6.1 (please see page29, line 468-488):

  Potential risks from inappropriately delivering PFA training was revealed from PFA trainees’ practice, such as confusion about role boundaries, inadequate skills development and inconsistent use of PFA guidance, which could potentially harm a person in distress [46,64]. Shortened PFA training led to insufficient understanding of the challenges in applying PFA [39]. PFA providers helping hurricane survivors reported that they often hesitated or over-estimated their roles in responding to survivors’ needs and felt they were unprepared to understand cultural differences [46]. Findings from a 1-day PFA training programme tailored to dealing with an outbreak of Ebola described potential risky practices in relation to the application of Hobfoll’s essential principles [64]. For example, giving a false reassurance to calm a distressed person by promising that everything would be fine which undermined their credibility and was counterproductive. Meanwhile, they also found trainees’ attempt to solve the problem for a distressed person rather than promoting the person’s own coping abilities, which could decrease self-efficacy and ultimately hinder recovery.

 The major strengths of PFA are that it is regarded as a low-cost intervention and is sufficiently simple that non-mental health workers can easily acquire PFA knowledge and skills, however, the limited training time, absence of post training supervision and lack opportunity for putting PFA skills into practice could all potentially reduce its efficacy. Horn et al. reported findings from large PFA training initiatives delivered during an Ebola outbreak which suggested that PFA knowledge and skills were not easily acquired for these reasons and should be treated with caution [64].

Point 15: Style

The grammar and wording within the paper reads awkwardly in places, and is misleading in others.  The paper would need to be checked for grammatical accuracy before publication.

Response 15: Thanks for your contribution to make this piece of work more clearly informed. We spent amount of time to improve our English expression by our British authors and have now again improved our writing accordingly.